# Fair Network Communities through Group Modularity

## Abstract

Communities in networks are groups of nodes that are more densely connected to each other than to the rest of the network, forming clusters with strong internal relationships. When nodes have sensitive attributes, such as demographic groups in social networks, a key question is whether nodes in each group are equally well-connected within each community. We model connectivity fairness through group modularity, an adaptation of modularity that accounts for group structures. We introduce two versions of group modularity grounded on different null models and present fairness-aware community detection algorithms. Finally, we provide experimental results on real and synthetic networks, evaluating both the group modularity of community structure in networks and our fairness-aware algorithms.

## CCS Concepts

• **Information systems** → **Data mining**.

## Keywords

algorithmic fairness, community detection, clustering, social networks, group modularity

**ACM Reference Format:**
Anonymous Author(s). 2018. Fair Network Communities through Group Modularity. In *Proceedings of Make sure to enter the correct conference title from your rights confirmation emai (Conference acronym 'XX)*. ACM, New York, NY, USA, 12 pages. https://doi.org/XXXXXXX.XXXXXXX

## 1 Introduction

Networks are essential for representing and analyzing interconnected systems across different domains, such as in social, collaboration, and citation settings. Nodes in networks often form communities, i.e., subsets of nodes that are more tightly connected with each other than with nodes outside the community [14, 24]. Connections in networks play a pivotal role in shaping opinions and influencing decision-making processes [13, 36]. In this paper, we study the *fairness* of connections within network communities.

Algorithmic fairness has been the center of much current research [11, 28, 30, 31]. In a broad sense, fairness is addressed either at the level of individuals, or at the level of groups of individuals [12, 33]. In most networks, nodes have attributes, forming groups, where nodes have the same values in one or more of their attributes. For example, in a social network, groups of nodes often correspond to demographic groups formed based on gender, age, or race. We consider fairness at the level of such groups.

Most previous research in group fairness of communities asks that the representation of groups within each community is balanced [7, 9, 23]. In this paper, we shift the focus from nodes to connections. We ask the key question, whether each group is equally well-connected within each community. For example, consider a collaboration network. Do women in the network participate in an equitable number of connections within the formed communities? The strength of connections within each community is vital for minorities to be heard, and influence others.

To model fairness of connections, we use modularity. Modularity is a measure of the quality of community structures in networks that quantifies the strength of the division of a network into communities by comparing the density of edges within communities to the expected density in a random graph [10, 29]. We introduce a variation of modularity, termed *group modularity*, that considers the density of edges of nodes belonging to a specific group. We consider two different random graph models. One agnostic to the group each node belongs to, and one that takes into account the group. In addition, we propose a diversity-based variation of modularity that looks only at connections between nodes belonging to different groups and we address its relationship to group fairness. Diversity of connections is important in addressing filter bubbles, and echo chambers, i.e., cases where individuals in a network are exposed only to opinions similar to their own often leading to reinforcing confirmation bias and polarization [13, 20, 27].

To locate fair community structures in a networks, we propose a fairness-aware community detection algorithm. The algorithm builds on the Louvain algorithm [6, 32], an agglomerative hierarchical method, where sets of nodes are successively merged to form larger communities such that modularity increases. In the proposed fairness-aware algorithms, the criterion for merging communities takes into account the fairness and diversity of the communities.

To evaluate our approach, we present experimental results using both synthetic and real networks. The goal of our experimental evaluation is multi-fold. First, we ask whether community structures are fair and diverse and what are the factors that affect fairness and diversity. Then, we evaluate the trade-off between the quality and the fairness and diversity of communities found by our community detection algorithms and compare the efficacy of the proposed models.

The remainder of the this paper is structured as follows. In Section 2, we introduce our model for fairness in communities, and in Section 3, we present the fairness-aware Louvain algorithm. Experimental results are reported in Section 4, related work in Section 5, while Section 6 concludes the paper.

## 2 Group-based Modularity Fairness

Let $G = (V, E)$ be an undirected graph, where $V$ is the set of nodes and $E \subseteq V \times V$ is the set of edges. We assume that nodes in $V$ belong to groups based on the value of one of their sensitive attributes.

We call red the protected value for this attribute. The red group, denoted by $R$, $R \subseteq V$, is the subset of nodes with red value. The blue group, denoted by $B$, $B \subseteq V$, $B \cup R = V$ and $B \cap R = \emptyset$, contains the remaining nodes. We will use $\phi$ to denote the ratio of the red nodes in the overall population, that is, $\phi = \frac{|R|}{|V|}$.

Assume that the nodes of the graph are partitioned into a set $C = \{C_1, C_2, \ldots C_k\}$ of $k$ communities. We will use $C_i^B$ and $C_i^R$ respectively for the blue and red nodes in community $C_i$.

Most previous research on group fairness focuses on node-based notions of community fairness [9, 23] that seek to maintain a balanced representation of the groups in each community, where the *red balance* of a community $C_i \in C$ is defined as: $B^R(C_i) = \frac{|C_i^R|}{|C_i|} - \phi$.

Given that network processes, including opinion formation, information propagation, and diffusion, primarily occur through interactions along the edges of the network [13, 36], in this paper, we look into group fairness from the edge perspective. To this end, we adopt a modularity-based approach.

*Modularity* measures the divergence between the number of intra-communities edges from the expected such number assuming a null model [10, 29]. The most commonly used null model is a random graph where the expected degree of each node within the graph is equal to the actual degree of the corresponding node in the real network. Specifically, the modularity of community $C_i$, $Q(C_i)$, is defined as [29]:

$$Q(C_i) = \frac{1}{2m} \left( \sum_{u \in C_i} \sum_{v \in C_i} A_{uv} - \frac{k_u k_v}{2m} \right) \quad (1)$$

where $A$ is the adjacency matrix of $G$, $m$ the number of edges in $G$ and $k_u$, $k_v$ the degree of node $u$, and $v$ respectively. Modularity provides a measure of how well nodes in a community are connected with each other. Negative values indicate less connections than expected, while positive values indicate more connections.

## 2.1 Group Modularity

Our goal is to ensure that red nodes are well connected within each community. Thus, for each red node $u$ in $C_i$ we take the difference between the actual number of its intra-community edges and the expected such number. We call this measure *red modularity*.

As before, the expected number of connections is estimated assuming as null model a random graph that preserves the degrees of nodes in $G$. Using this null model, red modularity, $Q^R(C_i)$ is defined as:

$$Q^R(C_i) = \frac{1}{2m} \sum_{u \in C_i^R} \sum_{v \in C_i} \left( A_{uv} - \frac{k_u k_v}{2m} \right). \quad (2)$$

We define similarly the *blue modularity* $Q^B(C_i)$. We refer to red and blue modularity collectively as *group modularity*.

Note that if we consider the whole graph as a single community both the red and blue modularity are zero. In general, positive values in a community mean that the nodes with the corresponding color are more connected in the community than expected.

We define *group modularity unfairness* by comparing the red and blue modularity:

*Definition 2.1.* For a community $C_i \in C$, the modularity unfairness of $C_i$, $u(C_i)$, is defined as:

$$u(C_i) = Q^R(C_i) - Q^B(C_i).$$

Negative values of $u(C_i)$ indicate unfairness towards the red group, that is, the fact that the red nodes are less well-connected within the community than the blue ones. Positive values indicate the opposite.

We also consider diversity within each community by considering the edges that connect nodes of different color, let us call these edges diverse edges,. Note that the expected number of diverse nodes cannot be estimated using the same null model, since to do so, we need to know the color of both endpoints of each edge. Thus, in this case, the expected number of diverse edges uses as null model a random bipartite graph with edges only between nodes of different color that preserves the degrees of the nodes in the original graph $G$. *Diversity modularity* ($D^{RB}$), or simply *diversity*, is defined as:

$$D^{RB}(C_i) = \frac{1}{2m} \sum_{u \in C_i^R} \sum_{v \in C_i^B} \left( A_{uv} - \frac{k_u k_v}{m} \right). \quad (3)$$

If we consider the whole graph as a single community, then diversity is a non positive value. In this case, the larger the value of $D^{RB}$ the more diverse the network.

*Simplifying the forms.* Let $X = \{R, B\}$ and $Y = \{R, B\}$. We use $In_i$ to denote the number of intra-community edges in $C_i$, $In_i^X$, the number of intra-community edges in $C_i$ with at least one endpoint belonging to group $X$, and $In_i^{XY}$ the number of intra-community edges with one endpoint in group $X$ and one endpoint in group $Y$. We also use $K_i$ for the sum of degrees of all nodes in $C_i$, and $K_i^X$ for the sum of degrees of all nodes in $C_i$ that belong to group $X$. With simple manipulations, we get:

$$Q^R(C_i) = \frac{1}{2m} \left( 2In_i^{RR} + In_i^{RB} - \frac{K_i K_i^R}{2m} \right) \quad (4)$$

$$u(C_i) = \frac{In_i^{RR} - In_i^{BB}}{m} - \frac{(K_i^R)^2 - (K_i^B)^2}{(2m)^2} \quad (5)$$

$$D^{RB}(C_i) = \frac{1}{2m} \left( In_i^{RB} - \frac{K_i^R K_i^B}{m} \right) \quad (6)$$

## 2.2 Labeled Group Modularity

We now consider a null model which is not agnostic of the color of edge endpoints. For a node $u$, let $k_u^R$ be the number of edges to red nodes and $k_u^B$ be the number of edges to blue nodes, $k_u^R + k_u^B = k_u$. In the following, $k_u^R$ and $k_u^B$ are respectively called the red degree and blue degree of node $u$.

We consider as null model a random graph where both the expected red degree and blue degree of each node within the graph is equal to the actual red degree and blue degree of the corresponding node in the real network $G$. Formally, let $P_{uv}$ be the probability of creating an edge between nodes $u$ and $v$. Let $m_{RR}$ be the number of red-red edges, $m_{RB}$ the number of red-blue edges and $m_{BB}$ the number of blue-blue edges. We have that $P_{uv} = k_u^R k_v^R / 2m_{RR}$, for red nodes $u, v \in R$, $P_{uv} = k_u^B k_v^B / 2m_{BB}$ for blue nodes $u, v \in B$, and

$P_{uv} = k_u^B k_v^R / m_{RB}$ for red-blue nodes $u \in R$ and $v \in B$. For any node $u$, it holds that $\sum_{v \in R} P_{uv} = k_u^R$ and $\sum_{v \in B} P_{uv} = k_u^B$.

We define the *labeled red modularity* $Q_L^R(C_i)$ by taking again the difference between the actual number of intra-community edges involving red nodes, and the expected such number, but now considering the color (or, in general, label) of both endpoints.

$$Q_L^R(C_i) = \frac{1}{2m} \left( \sum_{u \in C_i^R} \sum_{v \in C_i^B} \left(A_{uv} - \frac{k_u^B k_v^R}{m_{RB}}\right) \right. \tag{7}$$
$$\left. + \sum_{u \in C_i^R} \sum_{v \in C_i^R} \left(A_{uv} - \frac{k_u^R k_v^R}{2m_{RR}}\right) \right).$$

We define similarly the *labeled blue modularity* $Q_L^B(C_i)$. We refer to labeled red and labeled blue modularity collectively as *labeled group modularity*.

Again, if we consider the whole graph as a single community both the labeled red and labeled blue modularity are zero. In general, positive values in a community mean that the nodes with the corresponding color are more connected in the community than expected.

We define *labeled modularity unfairness* by comparing the red and blue labeled modularity.

*Definition 2.2.* For a community $C_i \in C$, the labeled modularity unfairness of $C_i$, $u_L(C_i)$, is defined as:

$$u_L(C_i) = Q_L^R(C_i) - Q_L^B(C_i).$$

Negative values of $u_L(C_i)$ indicate unfairness towards the red group, that is, the fact that the red nodes are less well-connected within the community than the blue ones. Positive values indicate the opposite.

We define *labeled diversity modularity*, or simply *labeled diversity*, as follows:

$$D_L^{RB}(C_i) = \frac{1}{2m} \left( \sum_{u \in C_i^R} \sum_{v \in C_i^B} \left(A_{uv} - \frac{k_u^B k_v^R}{m_{RB}}\right) \right). \tag{8}$$

In this case, the labeled diversity of the whole graph is zero. Positive diversity values in a community indicate that the community is more diverse than expected.

*Simplifying the forms.* We use $K_i^{XY}$, with $X \in \{R, B\}$, to denote the sum of the degrees of all nodes of color $X$ that belong to $C_i$ to any node of color $Y$. With simple manipulations, we get:

$$Q_L^R(C_i) = \frac{1}{2m} \left( 2 In_i^{RR} + In_i^{RB} - \frac{K_i^{RB} K_i^{BR}}{m_{RB}} - \frac{(K_i^{RR})^2}{2m_{RR}} \right) \tag{9}$$

$$u_L(C_i) = \frac{In_i^{RR} - In_i^{BB}}{m} - \frac{(K_i^{RR})^2 - (K_i^{BB})^2}{4\,m\,m_{RB}} \tag{10}$$

$$D_L^{RB}(C_i) = \frac{1}{2m} \left( In_i^{RB} - \frac{K_i^{RB} K_i^{BR}}{m_{RB}} \right) \tag{11}$$

*Discussion.* Note that both diversity and labeled diversity are symmetric, that is, it holds that $D^{RB}(C_i) = D^{BR}(C_i)$, and $D_L^{RB}(C_i) = D_L^{BR}(C_i)$.

Also, communities whose edges are all diverse have zero modularity unfairness and zero labeled modularity unfairness, e.g., they are fair under both definitions of unfairness. However, the opposite does not necessarily holds, that is, there be communities that are fair but not diverse.

## 3 Fairness-Aware Community Detection

In this section, we present our fairness-aware community detection algorithm. Our algorithm is based on the well-known Louvain algorithm that identifies communities in networks by optimizing modularity [6, 10, 32].

The algorithm follows a hierarchical agglomerative approach, starting with each node forming its own community. The original Louvain algorithm joins together two communities whose merge produces the largest increase in modularity $Q$ (Eq. 1). The fairness-aware algorithm uses two-criteria: two communities are joined if (1) modularity increases and (2) a group fairness criterion (*FC*) is met. For *FC*, we consider different approaches using either the non-labeled and the labeled group modularity, namely: (a) the *fairness-gain* approach where we ask that unfairness decreases (Eq. 5, or 10), (b) the *group-increase* approach, where we ask that the group modularity of the group towards which the network is unfair increases (Eq. 4, or 9), and (c) the *diversity-increase* approach where we ask that diversity increases (Eq. 6, or 11).

The algorithm works in two phases that are repeated iteratively. In the first phase, the algorithm computes for each node $u$ the gain in modularity and *FC* when removing $u$ from its current community and placing it to each of its neighboring communities. This process is applied repeatedly and sequentially and stops when a local maxima is reached, i.e., when no individual move can increase both modularity and *FC*.

In the second phase, the algorithm constructs a new graph whose nodes are now the communities found during the first phase. The weights of the edges between two nodes are the sum of the weights of the edges in the corresponding two communities. Edges between the nodes inside each community are modeled with a self-loop whose weight is the sum of the weights of these edges. Once the second phase completes, the first phase may be now applied to the resulting weighed graph, and then followed by the second phase. The iterations continue until there are no more changes and a modularity maximum is attained.

We assume that each edge $e$ is associated with a weight, $w(e)$. Thus, $k(u) = \sum_{v,(u,v) \in E} w(u,v)$, $k^X(u) = \sum_{(u,v) \in E, v \in X} w(u,v)$ and correspondingly the adjacency matrix is weighted. The following lemma (proof in the Appendix) estimates the change in red modularity $\Delta Q_{u \to C_i}^R$, blue modularity $\Delta Q_{u \to C_i}^B$ and diversity $\Delta D_{u \to C_i}^{RB}$ when an isolated red node moves to a community.

LEMMA 3.1. *When an isolated red node $u \in R$ is moved to community $C_i$, the difference $\Delta Q_{u \to C_i}^R$ in red modularity is:*

$$\Delta Q_{u \to C_i}^R = \frac{1}{2m} \left( 2 \sum_{v \in C_i, v \in R} w(u,v) + \sum_{v \in C_i, v \in B} w(u,v) - \frac{k_u(K_i + K_i^R)}{2m} \right)$$

the difference $\Delta Q^B_{u \to C_i}$ in blue modularity is:

$$\Delta Q^B_{u \to C_i} = \frac{1}{2m} \left( \sum_{v \in C_i, v \in B} w(u,v) - \frac{k_u K^B_i}{2m} \right)$$

and the difference $\Delta D^{RB}_{u \to C_i}$ in diversity is:

$$\Delta D^{RB}_{u \to C_i} = \frac{1}{2m} \left( \sum_{v \in C_i, v \in B} w(u,v) - \frac{k_u K^B_i}{m} \right)$$

Similar formulas hold when moving a blue node to $C_i$.

The following lemma (proof in the Appendix) estimates the change in labeled red $\Delta Q^R_{L,u \to C_i}$, labeled blue modularity $\Delta Q^B_{L,u \to C_i}$ and labeled diversity $\Delta D^{RB}_{L,u \to C_i}$ when an isolated red node moves to a community.

LEMMA 3.2. *When an isolated red node $u \in R$ is moved to community $C_i$, the difference $\Delta Q^R_{L,u \to C_i}$ in labeled red modularity is:*

$$\Delta Q^R_{L,u \to C_i} = \frac{1}{2m} (2 \sum_{v \in C_i, v \in R} w(u,v) + \sum_{v \in C_i, v \in B} w(u,v)$$
$$- \frac{k^B_u K^{BR}_i}{m_{RB}} + \frac{k^R_u K^{RR}_i}{m_{RR}})$$

the difference $\Delta Q^B_{L,u \to C_i}$ in labeled blue modularity is:

$$\Delta Q^B_{L,u \to C_i} = \frac{1}{2m} \left( \sum_{v \in C_i, v \in B} w(u,v) - \frac{k^B_u K^{BR}_i}{m_{RB}} \right)$$

and the difference $\Delta D^{RB}_{L,u \to C_i}$ in labeled diversity is:

$$\Delta D^{RB}_{L,u \to C_i} = \frac{1}{2m} \left( \sum_{v \in C_i, v \in B} w(u,v) - \frac{k^B_u K^{BR}_i}{m_{RB}} \right)$$

Similar formulas hold when moving a blue node to $C_i$. In the Appendix, we also present formulas applicable when merging communities.

## 4 Experiments

The goal of our experiments is to address the following research questions (RQ):

RQ1 What are the characteristics of a network that contribute to unfairness and lack of diversity within communities?

RQ2 Under which network conditions and through what modifications of fairness-aware Louvain algorithms can improvement in both notions of fairness be attained?

RQ3 How effective are the two definitions of unfairness and diversity in quantifying their respective measure and consecutively in improving fair community detection?

To address these questions we conducted experiments on both synthetic and real datasets.

### 4.1 Datasets

*4.1.1 Synthetic Datasets.* To study the factors that may lead to unfairness, we use a model based on the stochastic block model [19, 23] to create networks with nodes of different colors and connectivity behavior. The model has three important parameters: (1) Parameter

---

**Algorithm 1** Fairness-Aware Louvain

**Input:** Graph $G(V, E, A)$ where $V$ is the set of vertices, $E$ is the set of edges and $A$ is the attributes of each node in the graph
**Output:** List of $N$ clusters detected.
**repeat**
  **Assign** every vertex $v \in V$ to a singleton community
  **Calculate** the modularity $Q$
  **for** each vertex $v \in V$ **do**
    **for** each $u$ in neighbors of $v$ **do**
      **Calculate** the modularity gain $\Delta Q$ and fairness criterion change $\Delta Q_{FC}$ from the removal of $v$ from its current community and placement in the community of each neighbor.
      **if** modularity increases and $FC$ is met **then**
        Move $v$ to neighboring community
      **end if**
    **end for**
  **end for**
  **Create** a new "super-nodes" from the communities found on previous step. The new $V$ set is those "super-nodes".
  **Recalculate** the weight of the edges between these new "meta-nodes".
**until** there is no improvement

---

$\phi$ controls the size imbalance between the different groups. In a perfectly node-balanced network, $\phi = 0.5$; smaller values make the red group the minority one. (2) Parameter $p_c$ controls the probability of intra-community edges. In a random network with no community structure, $p_c = 0.5$; communities appear as $p_c$ increases. (3) Parameter $p_h$ controls the probability of same color edges, i.e., homophily. Values of $p_h$ larger than 0.5 result in homophily, while smaller than 0.5 result in heterophily. When $p_h = 0.5$, we have neutrality.

We start by an initial assignment of nodes in $k$ communities and then generate edges between the nodes. Note that the actual number of communities created may differ from $k$, depending on the values of the other parameters. An edge $(u,v)$ is generated with probability $p(u,v)$ defined as follows:

$$p(u,v) = \begin{cases} p_c \, p_h, & \text{if } u \text{ and } v \text{ are in the same cluster} \\ & \text{and have the same color} \\ (1-p_c) \, p_h, & \text{if } u \text{ and } v \text{ are in different clusters} \\ & \text{and have the same color} \\ p_c \, (1-p_h), & \text{if } u \text{ and } v \text{ are in the same cluster} \\ & \text{and have different colors} \\ (1-p_c) \, (1-p_h), & \text{if } u \text{ and } v \text{ are in different clusters} \\ & \text{and have different colors} \end{cases}$$

We also consider an asymmetric case, where we have different homophily probabilities, $p^R_h$ and $p^B_h$, for the red and the blue nodes respectively. When generating edge $p(u,v)$, we use $p^R_h$ if $u \in R$, and $p^B_h$ if $u \in B$.

Table 1 summarizes the parameters. We study the influence of size imbalance ($p_R$) and homophily ($p_h$). In each case, we vary one

**Table 1: Synthetic Dataset Characteristics**

| Parameter | Meaning | Default |
|---|---|---|
| $N$ | Number of nodes | 1000 |
| $\phi$ | Ratio of red nodes | 0.5 |
| $l$ | Avg node degree | 5 |
| $k$ | Initial number of communities | 5 |
| $p_h, p_h^R, p_h^B$ | Homophily | 0.5 |
| $p_c$ | Prob. of intra-community edge | 0.9 |

of the parameters and use the default values for the other. We run each experiment 10 times and report average values.

*4.1.2 Real datasets.* We study the following real datasets:

- **Pokec**[1] Nodes are the users of the Pokec social network and edges are friendship relationships between them. We consider the gender attribute (**Pokec-g**) and the age attribute (**Pokec-a**) as sensitive attributes. For the age attribute, we remove nodes that have no value for this attribute, or the value was not a possible value for age. We create two groups based on whether the user is over 30 years old or not.
- **Deezer**[2] Nodes are Deezer users from European countries and edges are mutual follower relationships between them.
- **Facebook**[3] The dataset consists of friends list from Facebook. We consider the gender attribute **Facebook-g** and the concentration attribute **Facebook-c**. The concentration attribute corresponds to the users that have chosen a specialized field of study within their major.
- **Twitch**[4] Nodes are twitch users and edges are mutual follower relationship between them.

The network characteristics are summarized in Table 2. We also report homophily values that indicate the tendency of nodes to connect with nodes with similar attributes, in our case, with the same color. We report separately the homophily of the red nodes ($Rh$) and the homophily of the blue nodes ($Bh$). Red homophily ($Rh$) is computed as the ratio of the number of the actual edges connecting two red nodes and the expected number of such edges (estimated as $\phi^2$). $Rh > 1$ indicate homophily, while $Rh < 1$ heterophily (tendency to connect with nodes of the opposite color). Similarly, we compute the blue homophily ($Bh$) as the ratio of number of the actual edges between two blue nodes and the expected number of such edges (estimated as $(1 - \phi)^2$).

## 4.2 Evaluation Results

To evaluate our approach on synthetic datasets, we create both symmetric and asymmetric datasets. Our goal is to examine how different group distributions and homophily patterns impact the performance of our algorithms.

We created datasets based on three distinct values of $\phi$:

- Red minority $\phi = 0.2$, where the red group is underrepresented relative to the blue group.

- Balanced groups $\phi = 0.5$, where the red and blue groups are evenly represented.
- Red majority group $\phi = 0.8$, where the red group forms the majority.

In addition to the group ratio, we adjust the homophily parameter $p_h$, which controls the likelihood of nodes within the same group to connect. We explore values of $p_h$ in the range from 0.1 to 0.9, capturing from strongly heterophilic ($p_h = 0.1$) to strongly homophilic ($p_h = 0.9$) networks. We create symmetric datasets, where the same homophily is assigned to both groups, allowing us to evaluate networks where both groups follow similar connectivity patterns. We also create asymmetric datasets, where the homophily parameter between the red and blue groups is different. In this case, we fix the homophily of the red group to be neutral ($p_h^R = 0.5$), and we vary the homophily of the blue group $p_h^B$, from 0.1 (heterophilic) to 0.9 (homophilic).

To address RQ1, we apply the original Louvain algorithm on each of the synthetic networks; the results are shown in the first row of Figures 1 (symmetric) and 2 (asymmetric). Our analysis reveals significant correlation between unfairness and diversity and the homophily and group size of a network. Specifically, higher homophily is associated with increased unfairness and reduced diversity, as nodes tend to connect with nodes of the same type, which reinforces group isolation within communities.

The most favorable outcomes for both network fairness and diversity occur when group sizes are balanced, and homophily is moderate, allowing for more cross-group interactions (Figure 2(c)). The highest levels of unfairness are observed when both groups exhibit high homophily. As both red and blue homophily increase, the unfairness metric moves further from zero, which indicates that both groups are predominantly forming internal connections, leading to high segregation between groups.

In the cases of group size disparity, we observe distinct patterns based on the homophily levels of the minority and majority groups. When the minority group has neutral homophily, we observe that fairness declines as the majority group becomes more homophilic, with unfairness values becoming increasingly negative. On the contrary, when the larger group has neutral homophily we find that the unfairness metric remains closer to zero, and diversity remains relatively high across most homophily levels of the minority group.

This pattern is also evident when Louvain is applied on the real world networks The detected communities are evaluated on the proposed metrics for diversity and unfairness, with the results presented in Table 3. In particular, we find that networks with high groups size disparities, such as Pokec-a and Facebook-c, exhibit high levels of unfairness within the detected communities.

To investigate RQ2, we evaluate several fairness-aware versions of the Louvain algorithm, incorporating the concepts of fairness-gain, group-increase, and diversity-increase introduced in Section 3. The minority group-increase modifications proved consistently to be the most effective in identifying fairer communities. We use the prefix $L$ for methods that use the labeled group modularity. In this study, we assume that the minority group is represented by the red group, and for our experiments, we refer to the corresponding methods as Red and L-Red. Rows 2 and 3 of Figures 1 and 2 show the results of the Red and L-Red methods on the synthetic networks.

---

[1]https://snap.stanford.edu/data/soc-Pokec.html
[2]https://snap.stanford.edu/data/feather-deezer-social.html
[3]http://snap.stanford.edu/data/ego-Facebook.html
[4]https://snap.stanford.edu/data/twitch_gamers.html

**Table 2: Network characteristics, $\bar{K}^X$: average degrees, Rh (Bh): red (blue) homompily.**

| Network | #Nodes | #Edges | Attribute | #Red nodes | #Blue nodes | $\bar{K}^R$ | $\bar{K}^B$ | $\bar{K}^{RR}$ | $\bar{K}^{BB}$ | $\bar{K}^{RB}$ | Rh | Bh | $\phi$ |
|---|---|---|---|---|---|---|---|---|---|---|---|---|---|
| Pokec-g | 1,632,636 | 22,301,602 | Gender | 804,335 | 828,301 | 26.33 | 28.28 | 5.18 | 6.39 | 15.49 | 0.770 | 0.922 | 0.492 |
| Pokec-a | 1,632,636 | 22,301,602 | Age | 239,785 | 1,392,851 | 15.95 | 29.27 | 0.79 | 13.40 | 2.47 | 0.394 | 1.149 | 0.146 |
| Deezer | 28,281 | 92,752 | Gender | 12,538 | 15,743 | 6.34 | 6.73 | 1.41 | 1.96 | 2.79 | 0.972 | 1.07 | 0.443 |
| Facebook-g | 4,039 | 88,234 | Gender | 1,533 | 2,506 | 45.75 | 42.42 | 10.24 | 13.48 | 15.45 | 1.236 | 0.995 | 0.378 |
| Facebook-c | 4,039 | 88,234 | Education | 367 | 3,672 | 31.38 | 44.92 | 2.94 | 21.18 | 2.54 | 1.481 | 1.066 | 0.090 |
| Twitch | 168,114 | 6,797,557 | Maturity | 79,033 | 89,081 | 88.25 | 74.31 | 24.57 | 19.80 | 34.69 | 1.292 | 0.924 | 0.470 |

**Table 3: Communities formed by the original Louvain.**

| Network | Communities | Modularity | Unfairness | L-Unfairness | Diversity | L-Diversity |
|---|---|---|---|---|---|---|
| Pokec-g | 41 | 0.716 | -0.031 | -0.031 | 0.180 | 0.192 |
| Pokec-a | 39 | 0.713 | -0.589 | -0.589 | 0.050 | 0.055 |
| Deezer | 89 | 0.683 | -0.103 | -0.101 | 0.141 | 0.160 |
| Facebook-g | 16 | 0.834 | -0.167 | -0.175 | 0.152 | 0.181 |
| Facebook-c | 16 | 0.834 | -0.725 | -0.722 | 0.038 | 0.042 |
| Twitch | 23 | 0.420 | 0.001 | -0.004 | 0.043 | 0.090 |

**Table 4: Communities formed by the fairness-aware Louvain using the red gain (Red method).**

| Network | Method | Communities | Modularity | Unfairness | L-Unfairness | Diversity | L-Diversity |
|---|---|---|---|---|---|---|---|
| Pokec-g | Red | 58,369 | 0.695 | -0.019 | -0.019 | 0.178 | 0.191 |
| Pokec-a | Red | 179,082 | 0.616 | -0.490 | -0.490 | 0.052 | 0.056 |
| Deezer | Red | 2,945 | 0.593 | 0 | 0 | 0.146 | 0.162 |
| Facebook-g | Red | 204 | 0.818 | -0.149 | -0.157 | 0.154 | 0.181 |
| Facebook-c | Red | 1,462 | 0.592 | -0.480 | -0.479 | 0.040 | 0.043 |
| Twitch | Red | 5,396 | 0.394 | 0.005 | 0 | 0.042 | 0.087 |

Both methods achieve reduction in unfairness, but at the cost of an increase in the number of communities. This pattern is also observed in the real-world datasets (Tables 4 and 10), particularly in networks with high group size imbalance, such as Pokec-a. A similar trade-off is evident in Table 8 where the results of a Louvain modification that aims at enhancing community diversity are presented. While the algorithm improves diversity in most networks, such as Deezer (0.157) and Twitch (0.053), it also leads to a notable increase in the number of communities. This further underscores the challenge of balancing the goal of improving network fairness and maintaining cohesive network structures. Additional results with other methods are found in the Appendix.

For the final research question RQ3, we found that while all algorithms successfully improve their respective fairness objectives, they often lead to an increase in the number of communities (second and third rows of Figures 2 and 1). Notably, we find that the Red modification is more effective at decreasing unfairness and ensuring equitable distribution of edges across groups, particularly in high homophily networks with group size disparity. However this improvement comes at the cost of an increase in the number of communites and a reduction in modularity. In contrast, the L-Red modifications are more effective in preserving high modularity while limiting the number of communities, especially in cases of group size disparity. However, it achieves a more moderate improvement in fairness.

## 5  Related Work

Fairness in machine learning has received considerable attention [11, 28, 30, 31]. At a high-level, fairness models are distinguished based on whether fairness is addressed at the level of individuals or at the level of groups of individuals [12, 33]. In this paper, we study the specific problem of group fairness of communities in networks. Community detection is similar to the more general problem of clustering defined as the task of grouping a set of objects in clusters such that the objects in the same cluster are more similar to each other than to those in other clusters [21]. In the case of communities, nodes are grouped so that nodes inside each community, i.e, cluster, are more tightly connected with each other than with nodes outside the community [14]. Next, we place our work in the context of previous work on defining and ensuring fairness in community detection and clustering.

As opposed to our approach that defines fairness based on node connections, most group fairness definitions are based on balancing the representation of each group within each cluster. The balanced approach to fairness in clustering was introduced in the seminal work of fairlets [9] to ensure that each protected group must have approximately equal representation in each cluster. The approach has been extended along various directions, such as to support scalability and distributed processing [2, 5, 7], more than one protected group [4] and parametric fair representation [3]. Another model of fairness based on proportionality does not assume the existence of

**Table 5: Communities formed by the fairness-aware Louvain using the labeled red gain (L-Red method).**

| Network | Method | Communities | Modularity | Unfairness | L-Unfairness | Diversity | L-Diversity |
|---|---|---|---|---|---|---|---|
| Pokec-g | L-Red | 1,440 | 0.691 | -0.024 | -0.023 | 0.172 | 0.188 |
| Pokec-a | L-Red | 7,645 | 0.658 | -0.531 | -0.531 | 0.057 | 0.058 |
| Deezer | L-Red | 323 | 0.649 | -0.061 | -0.059 | 0.142 | 0.164 |
| Facebook-g | L-Red | 17 | 0.830 | -0.162 | -0.169 | 0.152 | 0.182 |
| Facebook-c | L-Red | 21 | 0.822 | -0.711 | -0.0707 | 0.039 | 0.043 |
| Twitch | L-Red | 526 | 0.384 | 0.004 | -0.015 | 0.022 | 0.084 |

**Table 6: Communities formed by the fairness-aware Louvain to improve diversity (various methods used).**

| Network | Method | Communities | Modularity | Unfairness | L-Unfairness | Diversity | L-Diversity |
|---|---|---|---|---|---|---|---|
| Pokec-g | Blue | 41,273 | 0.705 | -0.038 | -0.038 | 0.181 | 0.194 |
| Pokec-a | Red | 179,082 | 0.616 | -0.490 | -0.0490 | 0.052 | 0.057 |
| Deezer | Diversity | 5,366 | 0.567 | -0.063 | -0.062 | 0.157 | 0.173 |
| Facebook-g | Red | 204 | 0.818 | -0.149 | -0.157 | -0.154 | 0.182 |
| Facebook-c | Red | 1,462 | 0.592 | -0.480 | -0.479 | 0.041 | 0.043 |
| Twitch | Blue | 1,299 | 0.401 | -0.005 | -0.007 | 0.053 | 0.088 |

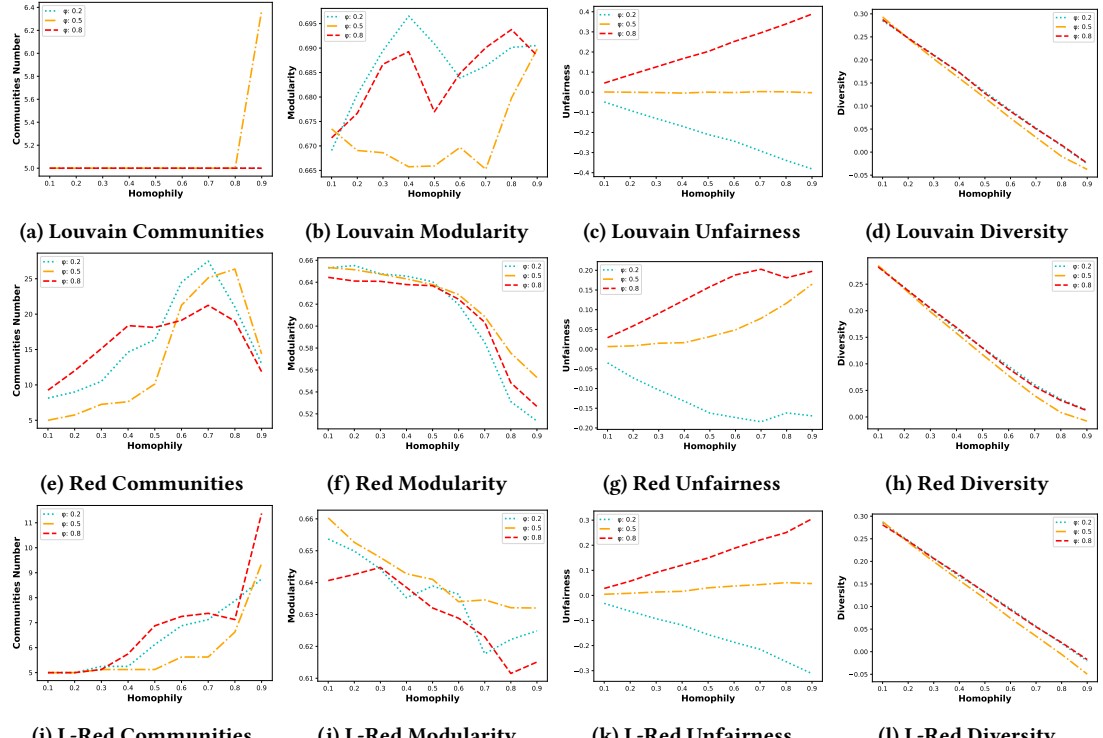

(a) Louvain Communities    (b) Louvain Modularity    (c) Louvain Unfairness    (d) Louvain Diversity

(e) Red Communities    (f) Red Modularity    (g) Red Unfairness    (h) Red Diversity

(i) L-Red Communities    (j) L-Red Modularity    (k) L-Red Unfairness    (l) L-Red Diversity

**Figure 1: Symmetric Datasets**

protected groups but seeks fair treatment for any subset of points [8]. Specifically, when clustering $n$ points with $k$ centers, any $n/k$ points are entitled to form their own cluster if there is another center that is closer in distance for all $n/k$ points. A related notion but for individual fairness was studied in [25] based on a previous formulation of the fair facility allocation problem.

A general definition of individual fairness in graph mining is that similar nodes should receive similar output. Applying this definition to graph clustering means that similar nodes should receive similar cluster assignments [22]. Given a similarity matrix that encodes the pair-wise similarity between nodes, this definition results in each node having most of its neighbors in this graph in the same cluster. The approach was extended in [35] for multiview graph clustering.

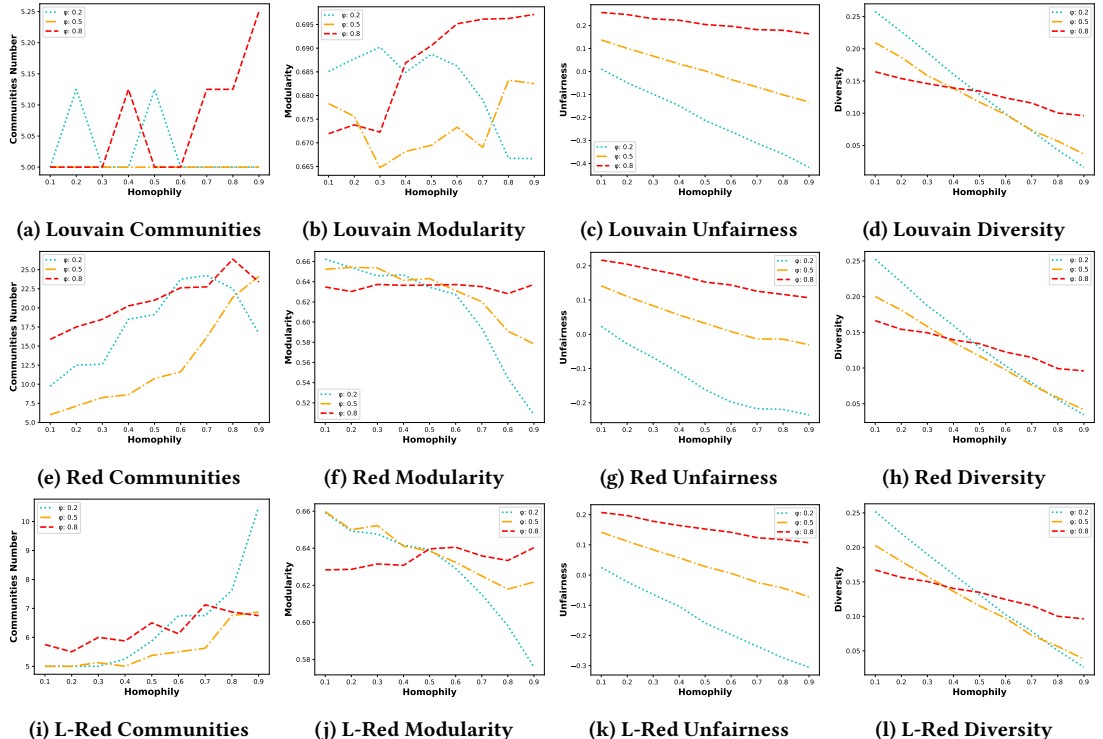

**Figure 2: Asymmetric Datasets**

Yet another approach assumes the existence of a representation graph between nodes and asks that the neighbors of each node are proportionately represented in each cluster [17, 18].

In terms of fairness definition in clustering, another view is to ask that the quality of clustering is the same for each group. This approach is taken in the *socially fair k*-means clustering approach that seeks to minimize the maximum of the average $k$-means objective applied to each group [15] and in *equitable* clustering that seeks to minimize the distance of each point to its nearest center [1]. In a sense, group modularity follows this view, since its goal is maintaining good clustering quality in terms of intra-cluster connectivity for each group.

In contrast to previous work, in this paper, we define community fairness through group modularity. Modularity has been refined to promote *mixed links*, i.e., links connecting nodes of different color in link recommendations [27]. This refinement is similar to our definition of diversity under the first null model. Red modularity using the first null model was also introduced in the short paper [26]. The other definitions of group fairness as well as the fairness-aware algorithms are novel in this paper.

Finally, in terms of algorithms for graph clustering, in this paper, we propose a modularity-based algorithm. Most previous work on addressing fairness considers algorithms based on spectral clustering. Most often fairness is achieved by adding fairness constraints to spectral clustering. Work in [23] adds constraints to spectral clustering for balancing nodes and is extended in [34] for scalability. Work in [22, 35] imposes individual fairness using spectral

clustering of the adjacency matrix combined with the similarity matrix. Spectral-based approaches are also followed in [16–18].

## 6 Conclusions

In many real-world networks, communities are formed, where nodes in each community are more tightly connected with the nodes inside their community than with the nodes outside their community. In this paper, we studied the fairness of such communities. Specifically, given that the nodes in a network belong to different groups, we examine whether the nodes of each group are equally well connected within the communities. To capture the fairness of communities, we proposed group modularity, an adaptation of the well known network modularity. We also used modularity to study the diversity of communities, i.e., the percentage of inter-group edges within each community. We proposed a fairness-aware Louvain-based algorithm that detects communities with good modularity, and fairness, or diversity. Our experimental evaluation showed the effects of homophily and size discrepancy in the fairness and diversity of the formed communities.

Our modularity-based metrics of fairness and diversity are orthogonal to the community detection algorithms used and as future work, we plan to investigate alternative approaches to community detection. Another direction for future work is understanding the evolution of community fairness and diversity through time and investigating approaches for improving the fairness and diversity of communities, for example, through appropriate link recommendations.

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

# A    Appendix

## A.1    Fairness-Aware Modularity detection

*Proof of Lemma 3.1.*

PROOF. Let $C_u$ be the single node community the red node $u$ belongs to before $u$ is moved to $C_i$. Moving $u$ only affects the modularity and diversity of $C_u$ and $C_i$.

Before moving $u$, it holds:

$$Q^R(C_u) = -\frac{1}{2m}\left(\frac{k_u^2}{2m}\right)$$

Before moving $u$:

$$Q^R(C_i) = \frac{1}{2m}\left(2In_i^{RR} + In_i^{RB} - \frac{K_i K_i^R}{2m}\right)$$

After moving $u$:

$$Q^R(C_i) = \frac{1}{2m}(2\frac{In_i^{RR} + 2\sum_{v\in C_i, v\in R} w(u,v) + In_i^{RB} + \sum_{v\in C_i, v\in B} w(u,v)}{m}$$
$$- \frac{(K_i + k_u)(K_i^R + k_u)}{2m})$$

Subtracting the before from the after values, we get $\Delta Q^R_{u\to C_i}$.

For the blue modularity of $C_u$, before moving $u$, we have $Q^B(C_u)$ is 0.

Before moving $u$:

$$Q^B(C_i) = \frac{1}{2m}\left(2In_i^{BB} + In_i^{BR} - \frac{K_i K_i^B}{2m}\right)$$

After moving $u$:

$$Q^B(C_i) = \frac{1}{2m}\left(2In_i^{BB} + In_i^{BR} + \sum_{v\in C_i, v\in B} w(u,v) - \frac{(K_i + k_u)K_i^B}{2m}\right)$$

Subtracting the before from the after values, we get $\Delta Q^B_{u\to C_i}$.
The diversity of $C_u$ before moving $u$ is 0.
Before moving $u$:

$$D^{RB}(C_i) = \frac{1}{2m}\left(In_i^{RB} - \frac{K_i^R K_i^B}{m}\right)$$

After moving $u$:

$$D^{RB}(C_i) = \frac{1}{2m}\left(In_i^{RB} + \sum_{v \in C_i, v \in B} w(u,v) - \frac{(K_i^R + k_u)K_i^B}{m}\right)$$

Subtracting the before from the after values, we get $\Delta D_{u \to C_i}^{RB}$.  □

*Merging Communities.* With similar manipulations, we get the red gain $\Delta Q_{C_i \to C_i}^R$. of merging two communities, community $C_i$ and community $C_j$.

Let $W_{XY}$ be the number of edges between nodes of color $X$ in community $C_i$ and nodes of color $Y$ in community $C_j$. Let $K_i$ be the sum of degrees of all nodes in community $C_i$, $K_i^X$ be the sum of the degrees of all node of color $X$ in community $C_i$ and $K_i^{XY}$ be the sum of the $Y$-colored degrees of the $X$-colored nodes in community $C_i$.

$$\Delta Q_{C_i \to C_j}^R = \frac{1}{2m}\left(2W_{RR} + W_{RB} - \frac{K_i K_j^R + K_j K_i^R}{2m}\right)$$

*Proof of Lemma 3.2.*

PROOF. Again, let $C_u$ be the single node community of the red node. Moving $u$ only affects the modularity and diversity of $C_u$ and $C_i$.

Before moving $u$, it holds:

$$Q_L^R(C_u) = -\frac{1}{2m}\left(\frac{(k_u^R)^2}{2m_{RR}}\right)$$

Before moving $u$, it holds:

$$Q_L^R(C_i) = \frac{1}{2m}\left(2\,In_i^{RR} + In_i^{RB} - \frac{K_i^{RB}K_i^{BR}}{m_{RB}} - \frac{(K_i^{RR})^2}{2m_{RR}}\right)$$

After moving $u$, it holds:

$$Q_L^R(C_i) = \frac{1}{2m}((2\,In_i^{RR} + In_i^{RB} + 2\sum_{v \in C_i, v \in R} w(u,v)+$$

$$\sum_{v \in C_i, v \in B} w(u,v) - \frac{(K_i^{RB} + k_u^B)K_i^{BR}}{m_{RB}} - \frac{(K_i^{RR} + k_u^R)^2}{2m_{RR}}$$

Subtracting the before from the after values, we get $\Delta Q_{L, u \to C_i}^R$.

The blue modularity $Q_L^B(C_u)$ of $C_u$, before moving $u$ is 0.

Before moving $u$, it holds:

$$Q_L^R(C_i) = \frac{1}{2m}\left(2\,In_i^{BB} + In_i^{BR} - \frac{K_i^{BR}K_i^{RB}}{m_{RB}} - \frac{(K_i^{BB})^2}{2m_{BB}}\right)$$

After moving $u$, it holds:

$$Q_L^R(C_i) = \frac{1}{2m}(2\,In_i^{BB} + In_i^{BR} + \sum_{v \in C_i, v \in B} w(u,v)$$

$$-\frac{K_i^{BR}(K_i^{RB} + k_u^B)}{m_{RB}} - \frac{(K_i^{BB})^2}{2m_{BB}})$$

Subtracting the before from the after values, we get $\Delta Q_{L, u \to C_i}^B$.

The diversity of $C_u$ before moving $u$ is 0.

Before moving $u$:

$$D_L^{RB}(C_i) = \frac{1}{2m}\left(In_i^{RB} - \frac{K_i^{RB}K_i^{BR}}{m_{BB}}\right)$$

After moving $u$:

$$D_L^{RB}(C_i) = \frac{1}{2m}\left(In_i^{RB} + \sum_{v \in C_i, v \in B} w(u,v) - \frac{(K_i^{RB} + k_u^B)K_i^{BR}}{m_{RB}}\right)$$

Subtracting the before from the after values, we get $\Delta D_{L, u \to C_i}^{RB}$.

□

*Merging Communities (labeled modularity).* With similar manipulations, we get the labeled red gain, $\Delta Q_{L, C_i \to C_i}^R$. of merging two communities, community $C_i$ and community $C_j$.

$$\Delta Q_{L, C_i \to C_j}^R = \frac{1}{2m}\left(2W_{RR} + W_{RB} - \frac{K_i^{RB}K_j^{BR} + K_i^{BR}K_j^{RB}}{m_{RB}} - \frac{K_i^{RR}K_j^{RR}}{m_{RR}}\right)$$

## A.2 Additional Experiments

This section presents the experimental evaluation of various fairness-aware modifications of the Louvain algorithm on both synthetic and real-world networks. The objective of these experiments is to assess the effectiveness of these modified algorithms in improving fairness and diversity within the detected communities, while preserving the overall quality of the community structure.

In Tables 7-12, we present results on the real datasets using additional fairness-aware Louvain algorithms.

In Figures 3 and 4 , we present results on synthetic datasets using the diversity gain and labeled diversity gain methods.

**Table 7: Communities formed by the fairness-aware Louvain using the red gain (Red method).**

| Network | Method | Communities | Modularity | Unfairness | L-Unfairness | Diversity | L-Diversity |
|---|---|---|---|---|---|---|---|
| Pokec-g | Red | 58,369 | 0.695 | -0.019 | -0.019 | 0.178 | 0.191 |
| Pokec-a | Red | 179,082 | 0.616 | -0.490 | -0.490 | 0.052 | 0.056 |
| Deezer | Red | 2,945 | 0.593 | 0 | 0 | 0.146 | 0.162 |
| Facebook-g | Red | 204 | 0.818 | -0.149 | -0.157 | 0.154 | 0.181 |
| Facebook-c | Red | 1,462 | 0.592 | -0.480 | -0.479 | 0.040 | 0.043 |
| Twitch | Red | 5,396 | 0.394 | 0.005 | 0 | 0.042 | 0.087 |

**Table 8: Communities formed by the fairness-aware Louvain using the diversity gain (Diversity method).**

| Network | Method | Communities | Modularity | Unfairness | L-Unfairness | Diversity | L-Diversity |
|---|---|---|---|---|---|---|---|
| Pokec-g | Diversity | 110,564 | 0.678 | -0.025 | -0.024 | 0.178 | 0.191 |
| Pokec-a | Diversity | 309,777 | 0.580 | -0.454 | -0.455 | 0.063 | 0.062 |
| Deezer | Diversity | 5,366 | 0.567 | -0.063 | -0.062 | 0.157 | 0.173 |
| Facebook-g | Diversity | 159 | 0.821 | -0.154 | -0.163 | 0.153 | 0.181 |
| Facebook-c | Diversity | 229 | 0.728 | -0.617 | -0.615 | 0.040 | 0.43 |
| Twitch | Diversity | 7,011 | 0.367 | 0.004 | 0 | 0.052 | 0.093 |

**Table 9: Communities formed by the fairness-aware Louvain using the fairness-gain gain (Fairness method).**

| Network | Method | Communities | Modularity | Unfairness | L-Unfairness | Diversity | L-Diversity |
|---|---|---|---|---|---|---|---|
| Pokec-g | Fairness | 42 | 0.713 | -0.031 | -0.031 | 0.179 | 0.179 |
| Pokec-a | Fairness | 36 | 0.709 | -0.586 | -0.586 | 0.049 | 0.055 |
| Deezer | Fairness | 68 | 0.681 | -0.102 | -0.101 | 0.140 | 0.140 |
| Facebook-g | Fairness | 17 | 0.823 | -0.166 | -0.171 | 0.146 | 0.179 |
| Facebook-c | Fairness | 12 | 0.794 | -0.688 | -0.684 | 0.034 | 0.041 |
| Twitch | Fairness | 26 | 0.422 | 0.002 | 0.001 | 0.039 | 0.091 |

**Table 10: Communities formed by the fairness-aware Louvain using the labeled red gain (L-Red method).**

| Network | Method | Communities | Modularity | Unfairness | L-Unfairness | Diversity | L-Diversity |
|---|---|---|---|---|---|---|---|
| Pokec-g | L-Red | 1,440 | 0.691 | -0.024 | -0.023 | 0.172 | 0.188 |
| Pokec-a | L-Red | 7,645 | 0.658 | -0.531 | -0.531 | 0.057 | 0.058 |
| Deezer | L-Red | 323 | 0.649 | -0.061 | -0.059 | 0.142 | 0.164 |
| Facebook-g | L-Red | 17 | 0.830 | -0.162 | -0.169 | 0.152 | 0.182 |
| Facebook-c | L-Red | 21 | 0.822 | -0.711 | -0.0707 | 0.039 | 0.043 |
| Twitch | L-Red | 526 | 0.384 | 0.004 | -0.015 | 0.022 | 0.084 |

**Table 11: Communities formed by the fairness-aware Louvain using the labeled diversity gain (L-Diversity method).**

| Network | Method | Communities | Modularity | Unfairness | L-Unfairness | Diversity | L-Diversity |
|---|---|---|---|---|---|---|---|
| Pokec-g | L-Diversity | 140,789 | 0.632 | -0.022 | -0.021 | 0.162 | 0.181 |
| Pokec-a | L-Diversity | 461,703 | 0.535 | -0.414 | -0.415 | 0.047 | 0.055 |
| Deezer | L-Diversity | 6,079 | 0.542 | -0.058 | -0.059 | 0.137 | 0.165 |
| Facebook-g | L-Diversity | 213 | 0.773 | -0.146 | -0.151 | 0.128 | 0.171 |
| Facebook-c | L-Diversity | 1,478 | 0.575 | -0.469 | -0.471 | 0.034 | 0.040 |
| Twitch | L-Diversity | 7,070 | 0.381 | 0 | -0.005 | 0.015 | 0.085 |

**Table 12: Communities formed by the fairness-aware Louvain using the labeled fairness-gain (L-Fairness).**

| Network | Method | Communities | Modularity | Unfairness | L-Unfairness | Diversity | L-Diversity |
|---------|--------|-------------|------------|------------|--------------|-----------|-------------|
| Pokec-g | L-Fairness | 34 | 0.713 | -0.031 | -0.032 | 0.179 | 0.191 |
| Pokec-a | L-Fairness | 36 | 0.714 | -0.591 | -0.591 | 0.051 | 0.055 |
| Deezer | L-Fairness | 98 | 0.687 | -0.104 | -0.102 | 0.140 | 0.162 |
| Facebook-g | L-Fairness | 15 | 0.834 | -0.167 | -0.175 | 0.152 | 0.181 |
| Facebook-c | L-Fairness | 16 | 0.834 | -0.725 | -0.722 | 0.038 | 0.042 |
| Twitch | L-Fairness | 22 | 0.423 | 0.001 | -0.001 | 0.042 | 0.091 |

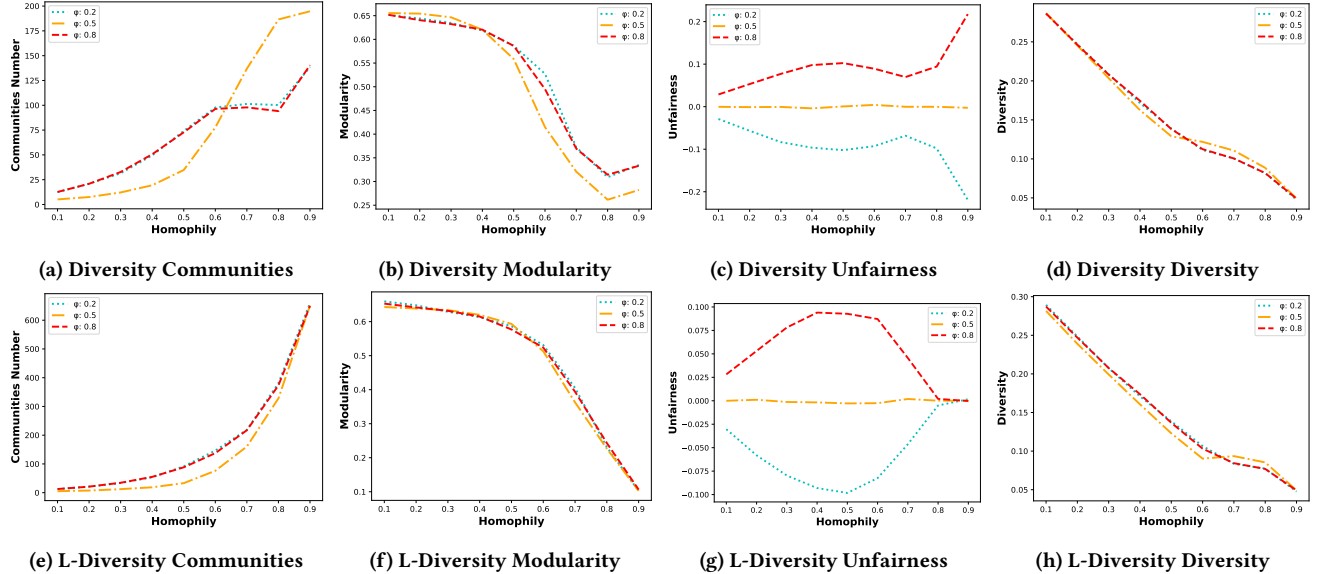

(a) Diversity Communities    (b) Diversity Modularity    (c) Diversity Unfairness    (d) Diversity Diversity

(e) L-Diversity Communities    (f) L-Diversity Modularity    (g) L-Diversity Unfairness    (h) L-Diversity Diversity

**Figure 3: Symmetric Datasets**

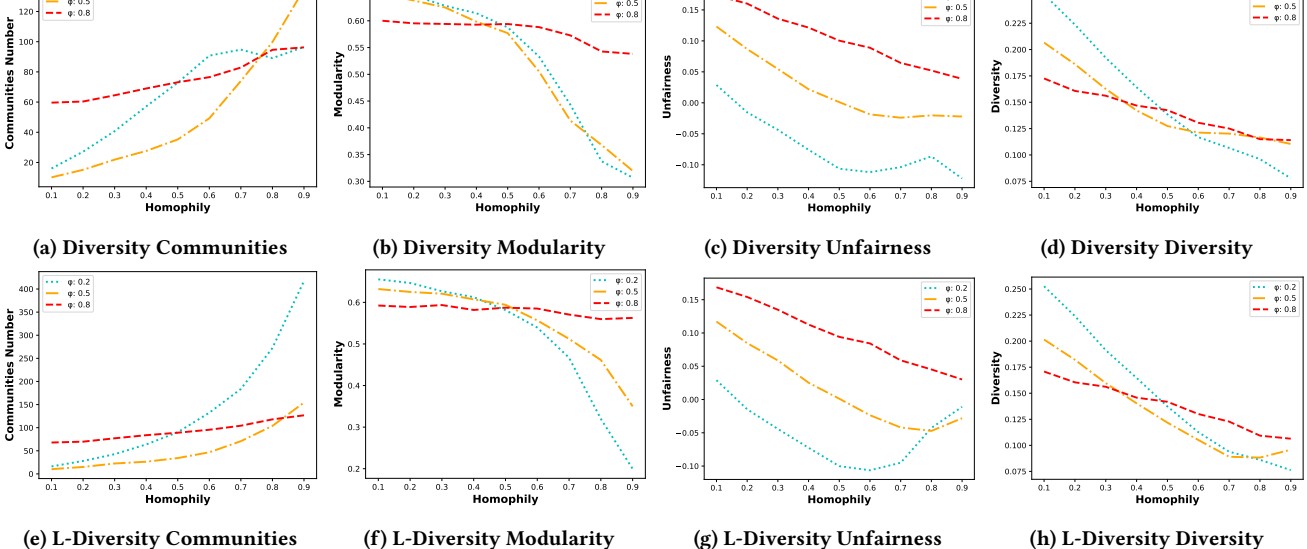

(a) Diversity Communities    (b) Diversity Modularity    (c) Diversity Unfairness    (d) Diversity Diversity

(e) L-Diversity Communities    (f) L-Diversity Modularity    (g) L-Diversity Unfairness    (h) L-Diversity Diversity

**Figure 4: Asymmetric Datasets**