# OpenReview forum: "Fair Network Communities through Group Modularity"
_ACM.org/TheWebConf/2025/Conference — WWW 2025 Oral_

### Official Review · Reviewer_sJfn · 2024-11-30

**Novelty:** 5
**Technical Quality:** 5

**Review:**

In this manuscript, the authors proposed several novel concepts regarding connectivity fairness and designed novel fairness-aware community detection algorithms. Although this manuscript is well-organized and of good quality, the clarity of this manuscript should be improved.

Pros：
1)	The concepts regarding connectivity fairness are relatively novel and interesting.
2)	The development of fairness-aware community detection algorithms is based on reasonable mathematical proof.
3)	Certain experimental results and conclusions are interesting and enlightening.
4)	Relatively complete experiments.

Cons:
1)	The range of certain proposed concepts, like the modularity unfairness, is still not clear enough.
2)	Certain concepts are not formalized well.
3)	When the authors illustrate their experimental findings, they do not clearly point out which specific figure and which specific line they are discussing. It is a bit difficult to follow the experimental parts of the manuscript.
4)	Certain conclusions of the experiments still need to be verified.
5)	Certain necessary theoretical computational complexity analyses are lacking.

**Questions:**

In the introduction, the authors mention that the key question is whether each group is equally well-connected within each community. I believe the“equally well-connected” here is a bit vague to understand, please formalize this concept.

For the definition, it seems like the Qr will change with the variation of Qb because the red group is the subset of nodes with red value and the blue group contains the remaining nodes. So, what is the range of the proposed modularity unfairness? Based on my knowledge, the range of the original modularity is [-0.5, 1]. The authors also need to point out the value of modularity unfairness when fairness is highest or the trend of modularity unfairness when fairness increases.

The diversity modularity also has the same problem. What is the range of the proposed diversity modularity?

The authors mention that the communities whose edges are all diverse have zero modularity unfairness and zero labeled modularity unfairness. This conclusion looks interesting, I believe that it is better to provide detailed proof to prove this conclusion.

Besides, the original modularity metric has the resolution limit problem, resulting in the modularity-optimization-based community detection algorithms having a limited ability to detect small communities. How do the authors solve this limitation?

Another limitation of the Louvain algorithm is the randomness problem. How do authors solve this question to ensure the final results are the same every time?

The authors should provide a theoretical computational complexity analysis of their Fairness-Aware Louvain method.


In experiments, authors mention that “when the larger group has neutral homophily we find that the unfairness metric remains closer to zero.” However, according to the red line of the figure 2.(c), the unfairness metric is much larger than 0. Besides, the authors also mention that “when the larger group has neutral homophily we find that the diversity remains relatively high across most homophily levels of the minority group.” However, according to the red, yellow, and blue lines of figure 2.(d), in around 40%-50% of cases, the diversity value of the blue line is much smaller than that of the red and yellow lines.


Based on the Table 3-6, it seems like the proposed method sacrifices the tightness of connections within the community to achieve fairness and diversity. I believe the trade-offs here need to be discussed in detail. For example, how much modularity is it reasonable to sacrifice to achieve diversity and fairness?

**Reviewer Confidence:**

3: The reviewer is confident but not certain that the evaluation is correct

**Scope:**

4: The work is relevant to the Web and to the track, and is of broad interest to the community

---

### Official Review · Reviewer_csv5 · 2024-12-02

**Novelty:** 4
**Technical Quality:** 4

**Review:**

Algorithmic fairness has become a central topic in recent research, often focusing on fairness at either the individual or group level. In many networks, nodes possess attributes that form groups, with members sharing common values in one or more attributes. For instance, in social networks, groups may be defined based on demographic factors such as gender, age, or race. This study examines fairness at the group level, particularly within the context of network communities.

Previous work on group fairness in communities has primarily emphasized balancing the representation of groups within each community. In contrast, this research shifts the focus from nodes to connections, posing the key question: are groups equally well-connected within their communities? For example, in a collaboration network, do women engage in an equitable number of connections within the communities? The strength of these connections is critical for ensuring minority voices are heard and influential within their communities.

To model the fairness of connections, the study employs modularity, a metric that evaluates the quality of community structures in networks by comparing the density of edges within communities to the expected density in a random graph. A variation termed group modularity is introduced, which accounts for the density of edges among nodes belonging to a specific group. Two random graph models are considered—one agnostic to group membership and another that incorporates it. Additionally, a diversity-based modularity variation is proposed, focusing on connections between nodes of different groups and exploring its implications for group fairness. Diversity of connections plays a crucial role in mitigating filter bubbles and echo chambers, where individuals are exposed only to similar opinions, reinforcing confirmation bias and polarization. The research emphasizes equitable connectivity and diversity in promoting fairness and reducing bias within network communities.

Experimentally, the authors consider three research questions:

RQ1 What are the characteristics of a network that contribute to
unfairness and lack of diversity within communities?
RQ2 Under which network conditions and through what modifica-
tions of fairness-aware Louvain algorithms can improvement
in both notions of fairness be attained?
RQ3 How effective are the two definitions of unfairness and di-
versity in quantifying their respective measure and consecu-
tively in improving fair community detection?

Although the paper offers some nice theory and methods, they are not, in themselves, enough for the conference. I'm not completely convinced by the experimental results, for example (see my questions below). The authors state in the introduction that this has been the center of much current research, but their own comparisons seem to be with classic methods. I would have liked to see a comparison to methods and baselines that are more state of the art. I'm hoping the authors can clarify this further for me.

**Questions:**

Can you comment on the statistical significance of the results? Also, I'm still not convinced that your results have been baselined properly. What are some alternative state of the art approaches to the problem you're considering, and have you compared with them?

**Reviewer Confidence:**

3: The reviewer is confident but not certain that the evaluation is correct

**Scope:**

3: The work is somewhat relevant to the Web and to the track, and is of narrow interest to a sub-community

---

### Official Review · Reviewer_QSye · 2024-12-02

**Novelty:** 5
**Technical Quality:** 5

**Review:**

This work addresses the issue of connectivity fairness within network communities, at a group level and by shifting the focus from nodes to edges.

The leading question is whether each group in a network is equally well-connected within its communities.
This scenario is modeled by relying on a variation of modularity (called $group$ $modularity$), capable of dealing with group structures within networks.

Based on the value of one of their sensitive attributes, nodes can belong to the red group $R$ (the group whose value for the attribute is protected), or to the blue group $B=V\setminus R$.

At this point, one can ask:

- whether nodes in $R$ are well represented in terms of connection within their community ($red$ $modularity$ and $group$ $modularity$ $unfairness$);

- whether these nodes are equally involved in diverse edges, i.e., edges connecting nodes of different colors ($diversity$).

The random graph used as null model can be either agnostic of the color of the endpoints of edges or not, meaning that for each of these three quantities, a ''labeled'' variant can also be considered.

After introducing these quantities, a ''fairness-aware'' community detection algorithm based on the Louvain algorithm is proposed:

- in the first phase, communities producing the largest increase in modularity are merged, with the additional requirement of meeting a ''fairness criterion'', i.e., increasing one of the above-mentioned group-based modularity quantities, either in its unlabeled or labeled variant;

- similarly to the Louvain algorithm, the second phase is devoted to building a weighted graph where nodes are communities and weights of edges are the sum of the weights of the edges between communities. The first phase is then repeated on this graph, and so on until the process stabilizes.

Three research questions are presented, and to answer them, an in-depth experimental evaluation follows.
Both synthetic and real-world networks with different structures and parameters (number of nodes, size imbalance between groups, homophily, ...) are used to perform the evaluation.

As one should expect, the experiments show that higher levels of homophily are correlated to more unfairness and less diversity, while decent fairness and diversity can be reached when the imbalance between groups and homophily are moderate.

The fairness-aware algorithm seems to perform better than the Louvain algorithm in terms of fairness and diversity, but in most cases, this is achieved by detecting a higher number of communities than those detected by the original algorithm.

***

- The paper is well-written, and the issue is described clearly, together with the proposed approach.

- The authors state clearly how and why their approach is different from the previous ones (i.e., they shift the focus from nodes to connections) but, at the same time, their proposal is an adaptation of something that is already well-known.

- The authors explicitly state what they want to evaluate in the experimental part, by posing three specific research questions. Moreover, the experiments include networks with a wide range of characteristics, and the different fairness criteria are evaluated extensively.

***

Some minor typos:

- Two paragraphs after Defition 2.1: a period punctuation immediately follows a comma.

- Last sentence of Section 2: a verb like ''might'' or ''can'' seems to be missing.

- Section 4.2, last paragraph before investigating RQ2: a period punctuation is missing before "The detected communities ...'' .

 - Title of Table 2: ''homophily'' is misspelled.

**Questions:**

- I don’t know if it is easily feasible, but a simple example showing how the proposed model differs from the original one (in terms of how the different notions of modularity, fairness, and diversity are assessed) could help grasp these differences.

- Are Table 4 and Table 7 the same table repeated twice? A similar issue involves Table 5 and Table 10.

- The fairness-aware algorithm does not seem to be unquestionably better than the Louvain algorithm at increasing fairness and diversity, mostly because the improvements are obtained at the cost of increasing the number of detected communities (RQ2 and RQ3). Have you thought about how to possibly deal with this issue, e.g. by adding some kind of penalization if the number of detected communities goes above some threshold, or would it excessively harm the performance of the algorithm?

**Ethics Review Description:**

No ethical issue with this paper has been noticed.

**Reviewer Confidence:**

3: The reviewer is confident but not certain that the evaluation is correct

**Scope:**

4: The work is relevant to the Web and to the track, and is of broad interest to the community

---

### Official Review · Reviewer_qwSg · 2024-12-03

**Novelty:** 6
**Technical Quality:** 5

**Review:**

In this paper, the authors model connectivity fairness through group modularity and aim to answer the key question of whether nodes in each group are equally well-connected within each community. The authors shift the focus of fairness research from the composition of nodes in communities (which has been the main concern in prior work) to the connections within communities. I think the paper is well-written, and the idea is novel. I generally like the paper. However, I find the measures introduced in this paper more interesting than the fairness-aware Louvain method. Perhaps the authors could improve their work by adding some applications for their method in the framing of the paper, but this is a minor suggestion. I have some comments/suggestions below.

**Questions:**

Comments/Questions:

- The definition assumes a binary split between red and blue groups. If there are more than two groups or if the sensitive attribute is continuous, the definition would need to be generalized. A brief discussion on this point would be helpful.

- The authors defined unfairness in Eq. 5 and Eq. 10 as a subtraction, but this approach has a potential issue: the desired value is around zero, and reducing unfairness can result in negative values, which still indicate deviations from fairness. Defining unfairness using absolute value signs would address this issue by treating both positive and negative deviations as equally unfair, ensuring clarity and consistency with the fairness objective. (I also think one reason for defining this is that they considered Red as the protected group, but the framing of the paper is more general, and possibly considering an absolute value makes more sense).

- The authors considered three fairness-aware community detection methods. These methods are not entirely distinct and appear to be interrelated. A brief discussion on how they are connected would add depth to the paper.

- On page 4, column 2, last line, the authors mentioned the size imbalance p_R; however, this was previously defined using \phi, which could be confusing. It would be clearer to use \phi consistently unless p_R refers to something different.

- Introducing briefly applications of the proposed algorithm would enhance the practical relevance of the work and provide additional context for its potential impact.

**Reviewer Confidence:**

4: The reviewer is certain that the evaluation is correct and very familiar with the relevant literature

**Scope:**

4: The work is relevant to the Web and to the track, and is of broad interest to the community

---

### Official Review · Reviewer_52cN · 2024-12-03

**Novelty:** 4
**Technical Quality:** 4

**Review:**

The paper introduces a framework for assessing connectivity fairness within network communities by adapting the concept of group modularity and proposes fairness-aware community detection algorithms, which are validated through experimental results on both real and synthetic networks. However, there are some issues with the  manuscript.
1. It doesn't adequately elaborate on the specific issues confronted by the current methodologies that the proposed solution intends to resolve.
2. It lacks comparison with existing baseline methods through experimental results.
3. There are some obvious grammatical errors present in the manuscript.
4. The content of the manuscript is not easy to follow.

**Questions:**

1.The manuscript mainly emphasizes the differences between the proposed method and existing ones, but what specific problems is this paper aiming to solve?
2. Why isn't there a comparison with existing baseline methods?

**Reviewer Confidence:**

3: The reviewer is confident but not certain that the evaluation is correct

**Scope:**

3: The work is somewhat relevant to the Web and to the track, and is of narrow interest to a sub-community